# Femtosecond laser programmed artificial musculoskeletal systems

Zhuo-Chen Ma[1,2], Yong-Lai Zhang [1✉], Bing Han[2], Xin-Yu Hu[1], Chun-He Li[1], Qi-Dai Chen[1] & Hong-Bo Sun [1,2✉]

Natural musculoskeletal systems have been widely recognized as an advanced robotic model for designing robust yet flexible microbots. However, the development of artificial musculoskeletal systems at micro-nanoscale currently remains a big challenge, since it requires precise assembly of two or more materials of distinct properties into complex 3D micro/ nanostructures. In this study, we report femtosecond laser programmed artificial musculoskeletal systems for prototyping 3D microbots, using relatively stiff SU-8 as the skeleton and pH-responsive protein (bovine serum albumin, BSA) as the smart muscle. To realize the programmable integration of the two materials into a 3D configuration, a successive on-chip two-photon polymerization (TPP) strategy that enables structuring two photosensitive materials sequentially within a predesigned configuration was proposed. As a proof-of-concept, we demonstrate a pH-responsive spider microbot and a 3D smart micro-gripper that enables controllable grabbing and releasing. Our strategy provides a universal protocol for directly printing 3D microbots composed of multiple materials.

---

[1] State Key Laboratory of Integrated Optoelectronics, College of Electronic Science and Engineering, Jilin University, 2699 Qianjin Street, Changchun 130012, China. [2] State Key Laboratory of Precision Measurement Technology and Instruments, Department of Precision Instrument, Tsinghua University, Haidian District, Beijing 100084, China. ✉email: yonglaizhang@jlu.edu.cn; hbsun@tsinghua.edu.cn

Recent years have witnessed the ground-breaking advances of microbots, which have revealed great potential for cutting-edge applications in microsurgery[1–3], cell manipulation[4–6], drug delivery[7–10], and biosensing[1,11–13]. Conventional microbots are mostly mechatronics systems that are composed of rigid materials (e.g., metals, silicon, and silica), and therefore suffer from poor biocompatibility, softness, flexibility, and biodegradability, which limits their applications in biomedical fields. Consequently, soft microbots that consist of soft and adaptable materials emerged as appealing alternative. Unlike hard-bodied robotic systems, soft microbots are generally based on smart materials with low Young's modulus, and thus feature high flexibility, biocompatibility and mechanical resilience to high loads[14–21]. Moreover, soft microbots and micro-motors can be actuated by a wide variety of external stimuli (e.g., electricity, magnetic field, light, heat, pH value, humidity, and chemical gradient), by which self-propelled motion, predictable deformation and controllable locomotion can be achieved without coupling additional energy-supply systems[22–34]. The aforementioned advantages make soft microbots promising for precise in vivo/ in vitro operations, for instance, human cell interaction, targeted drug delivery and minimally invasive surgery[2,4,5,22,35].

The advancement of soft microbots has been boosted by the rapid progress of material science[36,37] and micro/nanofabrication technologies[8,30,38,39]. A general approach to soft microbots is directly prototyping 3D micro/nanostructures based on smart materials. As typical examples, our group reported a dual-3D laser fabrication of soft actuators, in which both the 3D profile and the internal cross-linking gradient has been programmed during the two-photon polymerization (TPP) process[40]. Lewis et al. proposed a novel biomimetic 4D printing method and produced composite hydrogel configurations encoded with localized anisotropic swelling property through controllable alignment of cellulose fibrils along programmable 4D printing pathways[30]. Zhang et al. successfully fabricated magnetic microswimmer robots by coating magnetic materials on both artificial helical microstructures and natural biological matrix, enabling targeted gene delivery and in vivo imaging-guided therapy[2,41]. Wu et al. successfully fabricated conical hollow microhelices robots via femtosecond vortex beams generated by spatial light modulation[8]. The aforementioned successful examples have proven the importance of nanofabrication technologies for 3D soft microbots. Nevertheless, from the view point of materials, current strategies for developing soft microbots are generally based on a single stimuli-responsive material. Despite the fact that fast, dynamic and reversible deformation or locomotion can be realized in a controlled manner, concerns with respect to the isotropic response, weak mechanical strength for self-supporting, poor durability and robustness constitute the main barrier for practical applications.

Natural musculoskeletal systems provide the inspiration for developing robust yet flexible microbots that can combine the advantages of hard-bodied and soft microbots together. However, the fabrication of artificial musculoskeletal systems generally requires programmable assembly of two or more materials of distinct properties into complex 3D micro/nanostructures at submicron scale, which remains a big challenge. In this work, we report femtosecond laser programmed artificial musculoskeletal systems, in which relatively stiff photopolymer SU-8 serves as the skeleton structures, the soft and pH-responsive protein, bovine serum albumin (BSA), is employed as the smart muscle. The concept is enabled by successive on-chip TPP fabrication of the two photosensitive materials sequentially with the help of a microfluidic chip. As a proof-of-concept prototype of microbots, a pH-responsive spider microbot and a 3D smart gripper that enables controllable grabbing and releasing have been demonstrated. Our strategy may hold great promise for developing stimuli-responsive 3D microbots that are composed of multiple materials.

## Results

**On-chip TPP fabrication of musculoskeletal systems.** The essence of developing robotic musculoskeletal systems is the 3D integration of multiple materials at sub-micron scale. Nevertheless, currently available 3D fabrication strategies, for instance, 3D printing and stereolithography, are generally suitable for structuring only a solo material in a typical processing workflow. To realize multi-material integration within a 3D microstructure, it requires tedious yet sophisticated alignment of two or more materials during the fabrication process, including frequent exchange of material sources and highly precise positioning at nanoscale, which is almost impossible for real-case fabrication. Herein, to program the material composition of the 3D microstructures during the TPP fabrication, we developed a successive on-chip TPP strategy that enables in situ TPP, development, exchange of photoresist and multiple TPP integration. Figure 1a shows the scheme of our successive on-chip TPP system. A PDMS parapet is attached to the glass substrate, forming a microfluidic chip for flexible injection and emission of chemical reagents (e.g., photoresists and developer, Supplementary Fig. 1). In the first process, the 3D SU-8 skeleton of a spider was fabricated through a standard TPP process[42–45] according to a preprogrammed skeleton model. Then in situ developing process was implemented by injecting acetone to the chamber. In the secondary TPP fabrication, a stimuli-responsive material (Bull Serum Albumin, BSA) was injected into the chamber, and a successive TPP fabrication is initiated according to a muscle model that is well complementary to the skeleton model. After the removal of unpolymerized BSA gel, a 3D musculoskeletal system was fabricated. In this way, multiple materials that feature distinct properties can be integrated with in a 3D microstructure and the re-positioning process during TPP fabrication can be avoided.

To experimentally evaluate the capability of our successive on-chip TPP processing, we fabricated a proof-of-concept spider microbot using the aforementioned musculoskeletal system (SU-8 as skeleton and BSA as muscle). Figure 1b, c shows the SEM images of the micro-spider before and after muscle integration. The spider microbot consists of an SU-8 body and eight SU-8 legs that act as skeleton. At the junction of each leg, a vacancy is reserved for muscle integration. Notably, after the in situ alignment of BSA muscle, the entire spider configuration is completed. The optical microscopy images (insets of Fig. 1b, c) provide a direct comparison of the spider microbot before and after muscle integration. Due to the presence of the photoinitiator (methylene blue) in the BSA gel, the BSA muscles can be clearly identified. Detailed SEM images (Fig. 1d-f) indicate that the BSA muscles were integrated with the SU-8 legs exactly at the eight junctions (Supplementary Fig. 2). The interface between the skeleton and muscle is very smooth, indicating the high precision for multi-material integration. Actually, the bottom of the spider legs is detached to the surface of the substrate, as shown in the 3D model and the close-up SEM images (Supplementary Fig. 3). In aqueous solutions, these legs can be driven by varying the pH values of the surrounding medium. Figure 1g shows the actuation of the micro-spider by switching the pH value between 13 and 5. The expansion and the contraction of BSA muscles can lead to the bending and straightening of the legs (Supplementary Movie 1). To make this dynamic progress clear, the outlines of the spider with three different configurations are marked in red, yellow, and green, respectively. By overlapping these three profiles, the deformation of the legs can be compared clearly, whereas their SU-8 body seems unchanged. Such a smart

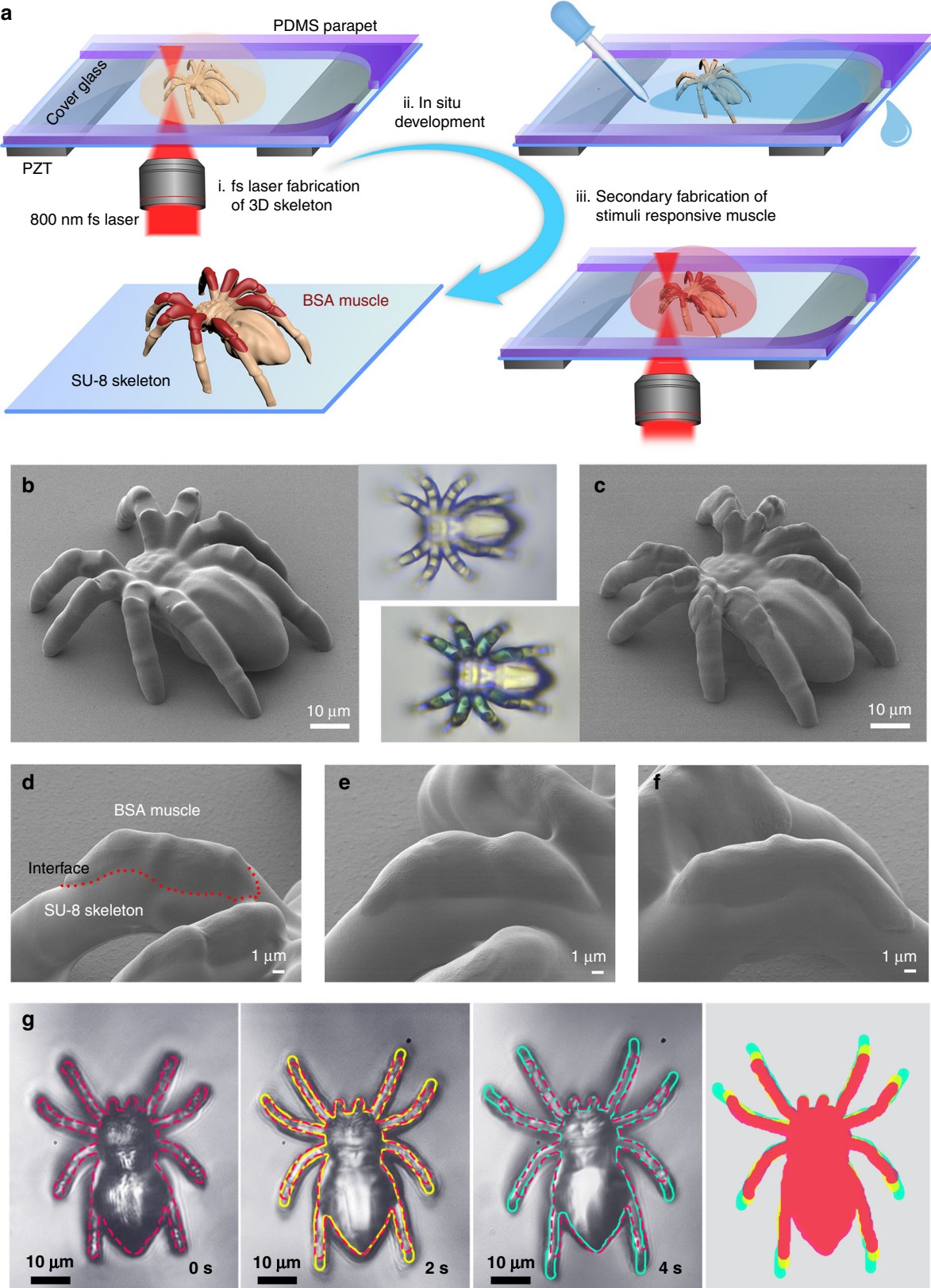

musculoskeletal system directly verifies the capability of successive on-chip TPP strategy for fabricating 3D microbots.

The used materials (SU-8, BSA, and their hybrid) were characterized using Fourier-transform infrared (FTIR) spectroscopy (Supplementary Fig. 4). Typically, the peaks at 3049, 1609, 758, and 659 cm$^{-1}$ in the spectrum of SU-8 (Supplementary Fig. 4a) correspond to the stretching vibration mode of aromatic =C–H, stretching vibration mode of aromatic C–C, out-of-plane bending mode of aromatic C–C, and out-of-plane bending mode of C–C. For BSA, the typical peaks at 3303, 1651, and 1536 cm$^{-1}$ correspond to the Amide A band (mainly N–H stretching vibration), Amide I band (stretching vibration of C=O), and

**Fig. 1 Scheme of the successive on-chip TPP strategy and a proof-of-concept spider microbot. a** Schematic illustration for the femtosecond laser programmable fabrication of the musculoskeletal systems. (i) TPP fabrication of the 3D SU-8 body and skeleton structure for eight legs; (ii) in situ development of the spider body and skeleton; (iii) secondary integration of the pH-responsive BSA muscles via TPP. **b, c** SEM images of the micro-spider before and after the integration of BSA muscles, respectively. The insets correspond to their optical microscopy images, and the blue parts in the latter one indicates the presence of BSA muscles at the joints of the legs. **d–f** Detailed SEM images of the joints where the BSA muscles were well integrated. **g** The actuation of the micro-spider when the pH value was switched from 13 to 5. The profiles of the micro-spider in this actuation are marked in red, yellow, and green, respectively. The superposition of the contour profiles (right image) at different time is provided for comparison.

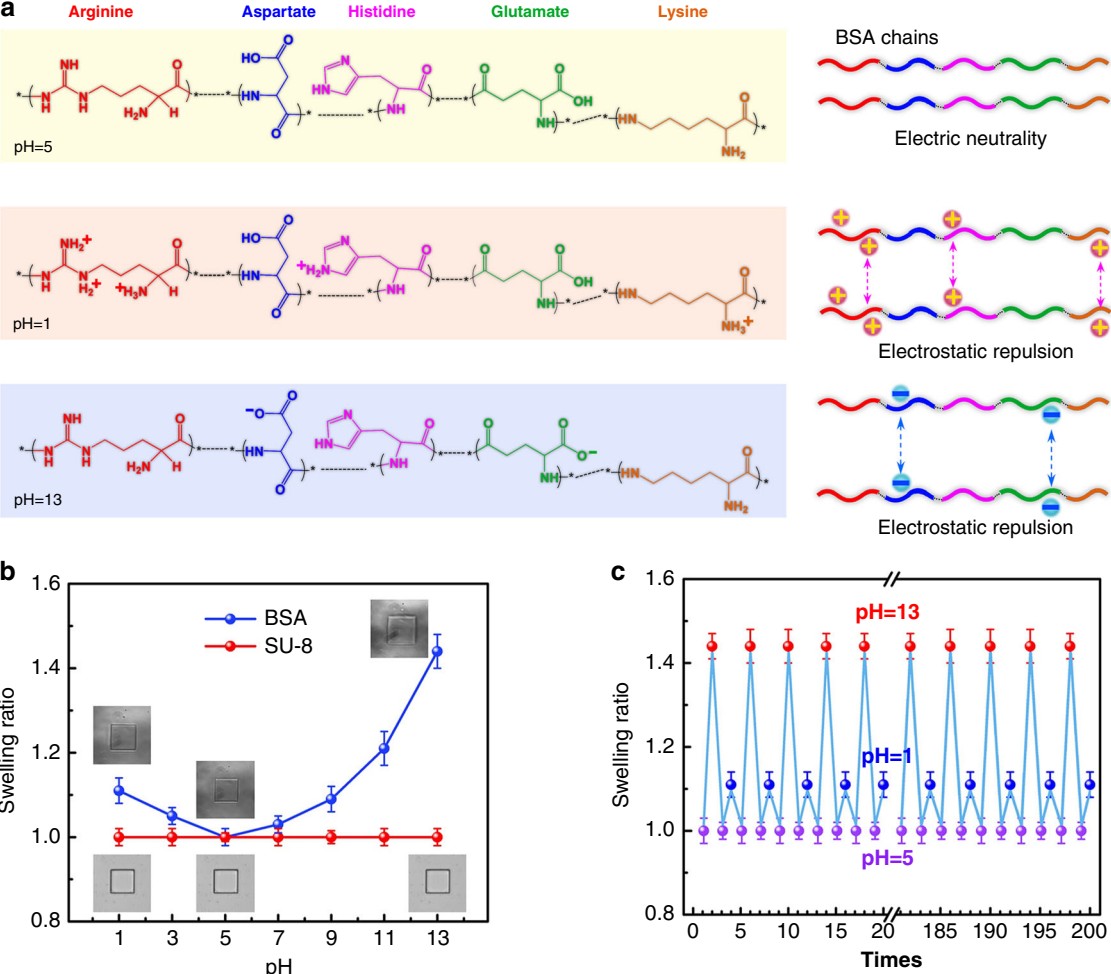

**Fig. 2 pH-responsive properties of the active BSA muscle and the inert SU-8 skeleton. a** Schematic illustration for the pH-responsive property of BSA molecular chains. Typical arginine, aspartate, histidine, glutamate, and lysine fragments were used for demonstrating their electrostatic interaction under pH values of 5, 1, and 13, respectively. At the isoelectric point (pH = 5), the BSA molecular chains are electrically neutral, thus exhibiting the smallest volume. At pH = 1, the amino groups of the BSA molecular chains are protonated, whereas the carboxyl groups are electrically neutral, so the molecular chains are positively charged. At pH = 13, the amino groups are electrically neutral, whereas the carboxyl groups are deprotonated, so the molecular chains are negatively charged. Due to the electrostatic repulsion of the BSA molecular chains, the volume of BSA structure increased. **b** The contrast of the swelling ratio of SU-8 and BSA blocks (10 × 10 μm in size). **c** Dynamic tuning of the size of BSA blocks by 200 times of pH switching. Error bars denote the standard deviation of the measurements. Source data are provided as a Source data file.

Amide II band (stretching vibration of C–N and bending of N–H), respectively (Supplementary Fig. 4b).

**pH-responsive properties.** In this study, the pH-sensitive protein BSA was employed as smart muscles for our musculoskeletal systems. Notably, the BSA structures fabricated via TPP demonstrated dynamic swelling and shrinking capability upon the variation of pH values of the surrounding medium[46–50]. Generally, the BSA hydrogel consists of two kinds of acidic amino acids (aspartate and glutamate) and three kinds of basic amino acids (arginine, histidine, and lysine) that dominate the volume change of BSA networks (Fig. 2a). The

isoelectric point of BSA is about 4.7; therefore, the BSA muscles show the smallest volume when the pH value is around 5. In acid condition (e.g., pH = 1), the amino groups of arginine, histidine, and lysine will be protonated and thus positively charged, whereas the carboxyl groups are electrically neutral. The strong electrostatic repulsion between the BSA molecular chains leads to the swelling of the BSA structure. When the surrounding medium is switched to basic solvents (e.g., pH = 13), the carboxyl groups of aspartate and glutamate will be deprotonated and thus negatively charged, whereas the amino groups are electrically neutral. The BSA structures also swell due to the strong electrostatic repulsion.

To make a quantitative comparison of the pH-responsive swelling behavior between BSA and SU-8, we investigated the swelling degree of BSA and SU-8 micro-squares (10 μm × 10 μm) under different pH values (Fig. 2b). In the case of BSA, the micro-square exhibited the smallest area at pH = 5. Its swelling ratio (defined as the area ratio of the micro-squire at a certain pH value to that at pH = 5) varies with the pH values, and the maximum swelling ratio of ~1.44 appears at pH = 13. On the contrary, the SU-8 structure keeps the same dimension throughout the pH range from 1 to 13, indicating the inertness under different pH conditions. The distinct pH responsiveness of BSA and SU-8 makes this musculoskeletal system a good candidate for developing pH-controlled 3D microbots. In addition, the swelling/shrinking behavior of the BSA under different pH values is stable and reversible. Figure 2c shows swelling ratio of the BSA structure during 200 times of pH switching in the order of pH = 5, 13, 5, and 1. Notably, the swelling ratio at a specific pH value keeps almost unchanged after repeated stimulation, indicating the stable responsiveness.

**Dynamic actuation of typical musculoskeletal systems**. As typical models of musculoskeletal systems, musculoskeletal actuators that were composed of BSA muscles and SU-8 skeletons were fabricated via successive on-chip TPP process; and their dynamic actuation performance was evaluated. Figure 3a shows the arm-muscle model. The skeleton is made of SU-8, in which the moving arm is about 35 μm in length and a vacancy is reserved at the joint for the BSA muscle. After the polymerization of the BSA muscle at the joint (Fig. 3a, middle images), an obvious straightening of the SU-8 arm can be observed, which can be attributed to the contraction of BSA under vacuum condition. The detailed SEM image of the two-material interface shows that the BSA muscle is tightly attached to the SU-8 skeleton, indicating the integrity of the musculoskeletal structure. The seamless integration of BSA and SU-8 is essential for the actuation performance. In principle, the two materials polymerize through different mechanisms. Thus, the BSA muscle may interact with SU-8 through a noncovalent manner. The FTIR spectrum of the SU-8/BSA hybrid (Supplementary Fig. 4) confirmed this issue, in which vibration peaks corresponding to newly formed covalent bonds cannot be detected. Besides, the arm-muscle system also demonstrates reasonable durability and robustness after long-time storage. The SEM images of the same structure after storage at 25 °C for 45 days demonstrate that it almost keeps the same morphology (Fig. 3a, bottom images). Figure 3b, c shows the actuation performance of the arm-muscle system. Dynamic folding and straightening performance can be observed when the surrounding pH value was switched from 5 to 13 (Supplementary Movie 2). The deformation of the device is very sensitive to the surrounding pH value, the structure folds from 2° to ~19° in 1.2 s upon the pH is changed from 5 to 13, and recovers to the initial shape (2°) within 1.5 s when the pH value is switched back to 5. Notably, after 45 days, the arm-muscle device shows comparable folding performance, indicating the good stability. Generally, the good durability can be attributed to the following two points. First, the stiff SU-8 structure can serve as the skeleton to support the whole musculoskeletal systems, so the device is robust enough for repeated actuation. Second, both SU-8 and the BSA hydrogel are chemically stable under proper storage conditions. The skeleton material SU-8 is a chemically inert resin that is very stable at room temperature, whereas the active muscle material BSA should be preserved in ultrapure water at low temperature to avoid biodegradation.

Figure 3d demonstrates another model, a crab claw-muscle system, in which a pair of zigzag pincers are made of SU-8 and

the BSA muscle is integrated at the joint of one pincer. Similar with the arm-muscle device, the BSA muscle can be identified clearly in the optical microscopy image. The shrinking and swelling of the BSA muscle under pH stimuli can cause the opening and closing of the SU-8 crab claw (Fig. 3e, Supplementary Movie 3). To investigate the dynamic actuation of the claw, we change the surrounding pH value between 5 and 13, and measure the folding angle of the pincers (Fig. 3f). In four consecutive cycles, this system exhibited dynamic, fast and reversible "open" and "clamped" behaviors. Actually, the musculoskeletal systems are not limited to the above-mentioned two models. By rational design of the musculoskeletal structures, our successive on-chip TPP fabrication enables directly prototyping of any desired 3D smart structures, revealing great potential for developing soft microbots.

**Smart micro-gripper**. To make full use of the musculoskeletal system, we designed and fabricated a pH-driven micro-gripper that consisted of the SU-8 skeleton and BSA muscle (Fig. 4a). The dimension and configuration of the micro-gripper before and after BSA integration are shown in Fig. 4b, c, respectively. Notably, the initial gripper skeleton was not in an open state. After the integration of BSA, the gripper becomes open due to the contraction of BSA muscles in air. Taking advantage of the programmable fabrication capability[44,45,51,52], the internal networks of the polymeric structure can be further controlled at nanoscale by varying the laser scanning step length. In this way, both the elasticity of BSA muscles and the stiffness of SU-8 skeletons can be precisely tuned. Figure 4d shows the dependence of the swelling ratio of a BSA micro-square (10 μm × 10 μm) on the step length when the pH value of the solution was switched between 5 and 13. With the increase of step length, the swelling ratio increased accordingly, as shown in the insets of Fig. 4d. The maximum swelling ratio of ~1.56 is achieved when the step length is 200 nm. Nevertheless, when the BSA blocks were integrated with the SU-8 micro-gripper as the muscles, the folding angle (depicted in the inset of Fig. 4e) of the micro-gripper did not follow the same tendency. The maximum folding angle of ~23° appears at the step length of 100 nm. Unlike the swelling performance in free space (Fig. 4d), the BSA muscles integrated with the micro-gripper were severely restricted by the SU-8 skeletons (the step length for SU-8 is 200 nm). Consequently, the folding of the micro-gripper is governed by the swelling force, rather than the swelling ratio. Generally, the BSA muscles fabricated under large step lengths show relatively poor mechanical strength for driving the SU-8 micro-gripper, whereas that fabricated under a small step length may show unobvious volume change. In this regard, the step length of 100 nm is an optimized parameter for the micro-gripper.

To make a direct comparison of the responsiveness of these micro-grippers, we fabricated four micro-grippers, in which the muscles were fabricated under the step length of 50, 100, 150, and 200 nm, respectively (Fig. 4f–i). At pH = 5, the BSA muscle undergoes a shrinking state, the four micro-grippers show different folding angles due to the difference of the shrinking force of the BSA. When the pH value is switched to 13, the BSA muscles swell, leading to the gripping behavior. The BSA muscle fabricated under the step length of 100 nm shows the best folding and unfolding performance (Fig. 4g, Supplementary Movie 4).

In addition to the BSA muscle, the stiffness of the SU-8 skeleton could also be altered by tuning the laser scanning step length. In a controlled experiment, we fixed the laser scanning step length of BSA at 100 nm, and varied the step length to 100 and 300 nm in the fabrication of SU-8 for comparison (Supplementary Fig. 5). The SU-8 skeleton fabricated under a step length of 100 nm is rigid, and a small folding angle of ~8° is observed (Supplementary Fig. 5a, b). When the step length was

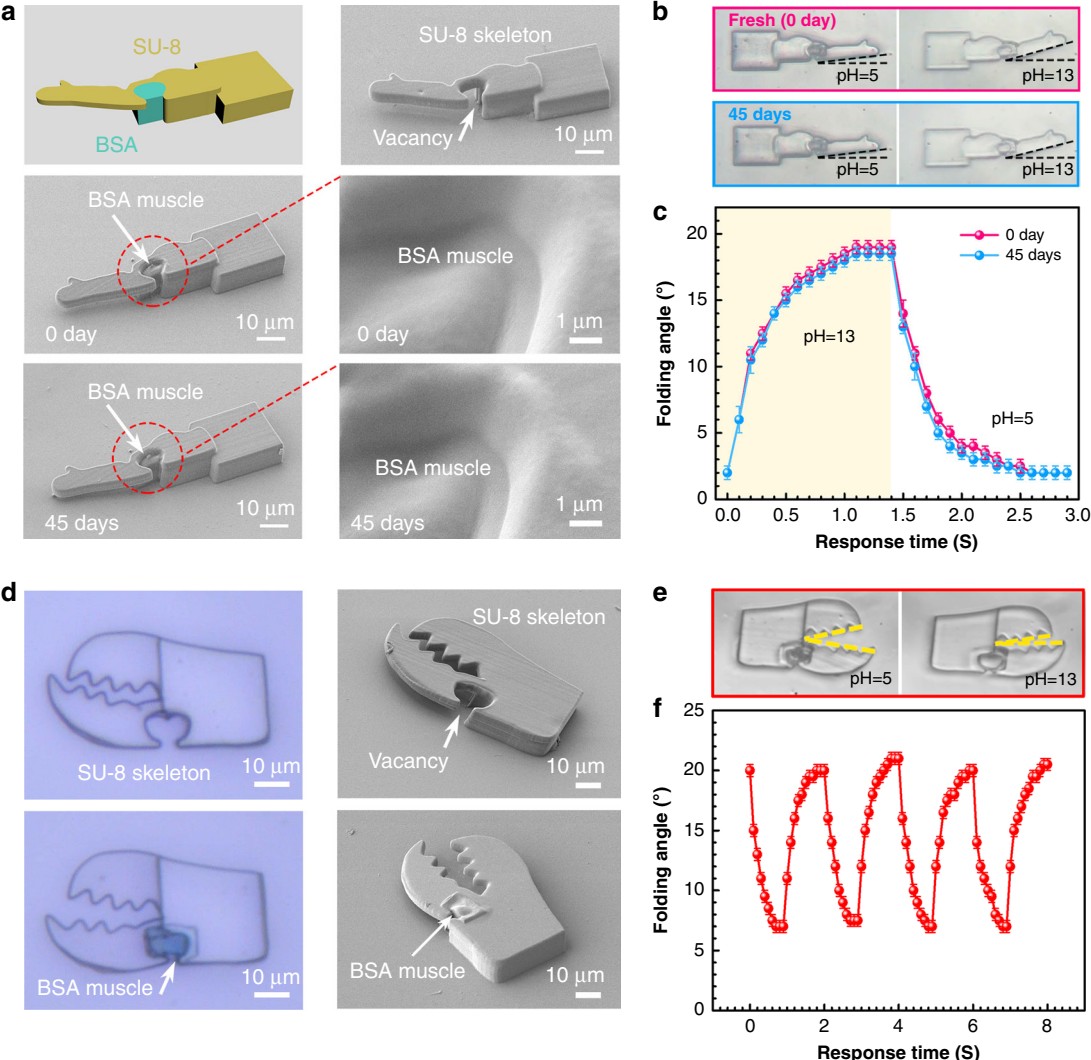

**Fig. 3 Dynamic pH actuation of two typical musculoskeletal systems. a** Arm-muscle system: Top images: the model (left) and SEM image (right) of the SU-8 skeleton. Middle images: SEM image (left) of the arm-muscle after BSA integration and SEM image of the BSA and SU-8 interface (right). Bottom images: SEM images of the arm-muscle (left) and the two materials interface (right) after storage for 45 days. **b** Optical microscopic images of the pH actuation of the arm-muscle and that after 45 days. **c** Dynamic pH-responsive properties of the arm-muscle structure. **d** The crab claw-muscle system: Top images: Optical microscopic image (left) and SEM image of the SU-8 skeleton (right); bottom images: Optical microscopic image (left) and SEM image of the claw after BSA integration (right). **e** Optical microscopic images of the pH actuation of the claw. **f** Dynamic pH-responsive properties of the crab claw for 4 cycles. Error bars denote the standard deviation of the measurements. Source data are provided as a Source data file.

increased to 300 nm, the stability of the micro-gripper decreased obviously (Supplementary Fig. 5e, f). It began to deform and even collapsed. To acquire a relatively large bending range and good stability, the step length of 200 nm is suitable for the SU-8 skeleton. The Young's modulus of the SU-8 microstructures fabricated under this condition was measured to be 4.8 GPa (Supplementary Fig. 6).

To evaluate the tension of BSA muscles exerted on the SU-8 skeletons, a pair of cantilevers (30 μm in length) was fabricated for analysis (Supplementary Fig. 7a). After integration of BSA muscles at the free ends of the two cantilevers (Supplementary Fig. 7b), the cantilevers can be driven by the BSA muscle upon pH change. The maximum displacement of the free end was measured to be 2.05 μm. Accordingly, a mathematical model based on approximate differential equation of the deflection curve was set up to analyze the tension produced by the contraction of the BSA muscle (Supplementary Fig. 7c). The tension was calculated to be 1.4–1.5 μN (see Calculation of the tension exerted by the BSA muscle on an SU-8 cantilever in the Supplementary

Materials for details). Besides, the deflection of the cantilever beam is also simulated using the COMSOL Multiphysics based on finite element analysis (Supplementary Fig. 8), in which the left end is fixed and the right end is subjected to a force $F$ (see Simulation of the deflection of the cantilever beam in the Supplementary Materials for details). When the applied force $F$ is 1.4–1.5 μN, the deflection of the end is 1.98–2.12 μm, which is consistent with the experimental results. Based on these experimental and theoretical results, the BSA muscle with 25 μm in length and 2 μm in width can produce a tensile force of 1.4–1.5 μN on the SU-8 cantilevers upon pH change.

The maximum weight that BSA muscles can drive depends on the size of BSA, the pH value for actuation and the laser processing parameters. To analyze the maximum weight driven by the BSA muscle, we fixed the physical parameters of the BSA muscle model (25 μm in length and 2 μm in width, laser scanning step length of 100 nm, laser power of 20 mW) and varied the widths of the SU-8 cantilevers within a certain range (4, 6, and 8 μm, Supplementary Fig. 9). According to the experimental

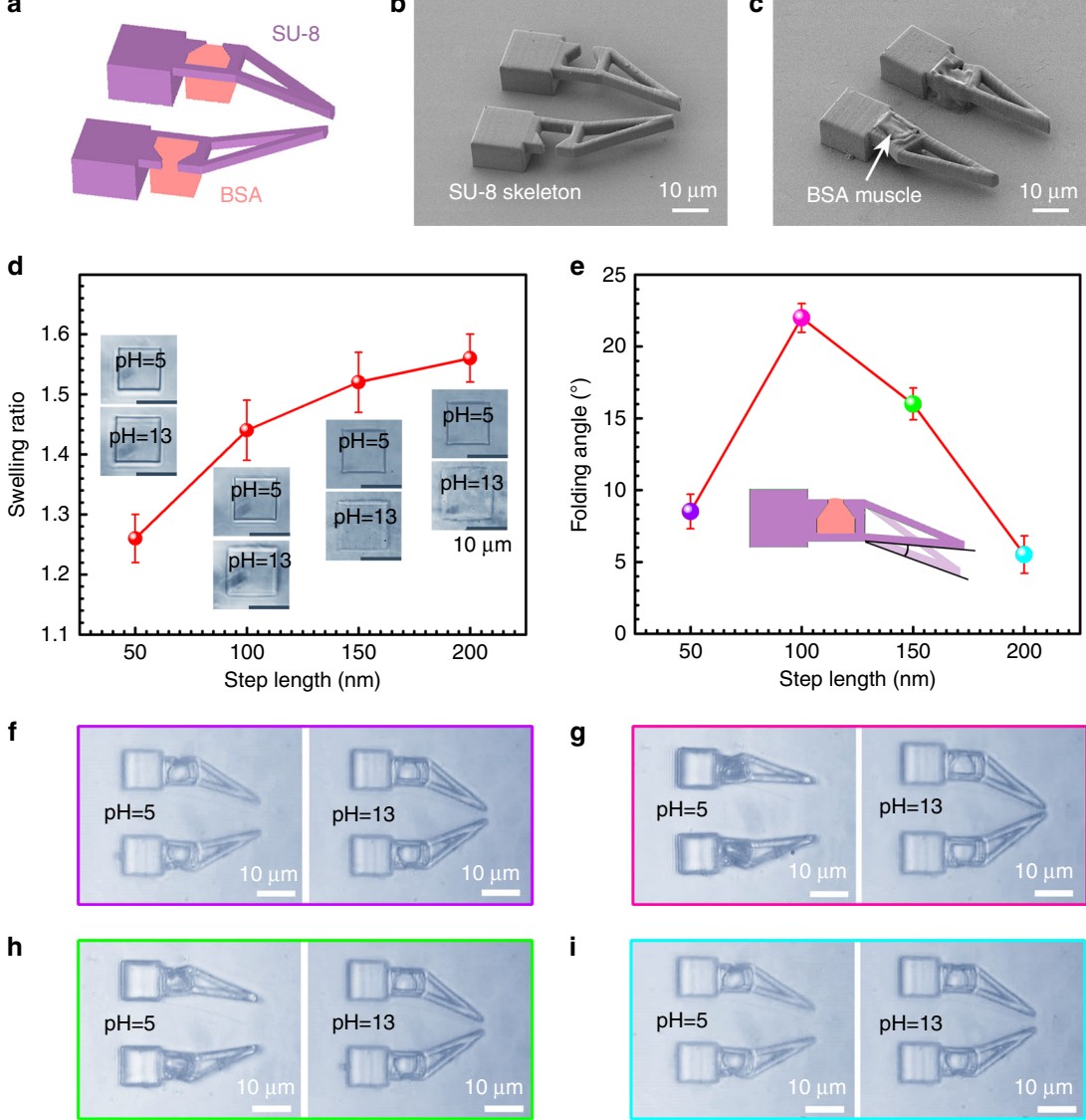

**Fig. 4 A pH-driven micro-gripper. a** The 3D model of the micro-gripper. **b, c** SEM images of the micro-gripper before and after the integration of BSA muscles, respectively. **d** Dependence of the swelling ratio of a BSA micro-block (10 μm × 10 μm) on the laser scanning step length (50, 100, 150, and 200 nm) when the pH of the surrounding solution was switched between 5 and 13. The insets are corresponding optical microscopic images of the BSA blocks. **e** The dependence of the folding angle (one arm of the micro-gripper) on the laser step lengths for the fabrication of BSA muscles. **f–i** The pH-responsive grapping performance of the micro-grippers with BSA muscles fabricated under the laser step lengths of 50, 100, 150, and 200 nm, respectively. Error bars denote the standard deviation of the measurements. Source data are provided as a Source data file.

results, the BSA muscle can drive the SU-8 cantilevers with a width of 6 μm, corresponding to the maximum force of ~34 μN. The SU-8 cantilever of 8 μm in width cannot be driven.

By controlling the folding and unfolding of the gripper via pH change, the gripper can be well manipulated to pick up and release targets of different sizes. Figure 5a shows the dependence of folding angle of one pincer on pH value. The micro-gripper presented the maximum and minimum folding angle (correspondingly the minimum and maximum spacing) when the pH is 13 and 5, respectively (Supplementary Fig. 10a, b). The dynamic actuation test shows that the gripper is very sensitive to pH change, it takes only 1.6 s to reach the maximum folding angle, and additional 2.3 s to recover to the original shape when the pH value was switched between 5 and 13 (Fig. 5b). If the surrounding pH value of the gripper was changed from 5 to others (e.g., 1, 3, 11, and 13), the folding angle could be flexibly tuned (Supplementary Fig. 11a). Since the laser programmed

musculoskeletal system is very robust, the gripper can undergo repeated opening and clamping in 100 response cycles (Supplementary Fig. 11b). In this process, the folding angle was tuned reversibly from ~23° to 0° when the pH value was varied between 13 and 5. Snapshots of the pH-driven micro-gripper show the dynamic gripping behavior; it deforms from the initial opening state to the clamping state within 2 s (Fig. 5c). To demonstrate the application of such a smart micro-gripper, we integrated the gripper with a glass cantilever that was subsequently fixed in a precision motion control system. A micro-cube (side length 10 μm) tied to a polymer base by a rope was employed as a target object (Fig. 5d). Figure 5e shows the schematic illustration of the manipulation procedures, including positioning, actuation, capturing, and releasing; and the snapshots of the process at different time is shown in Fig. 5f (Supplementary Movie 5). In the positioning process, the micro-gripper is manipulated to approach the target with the help of a 3D moving stage. When

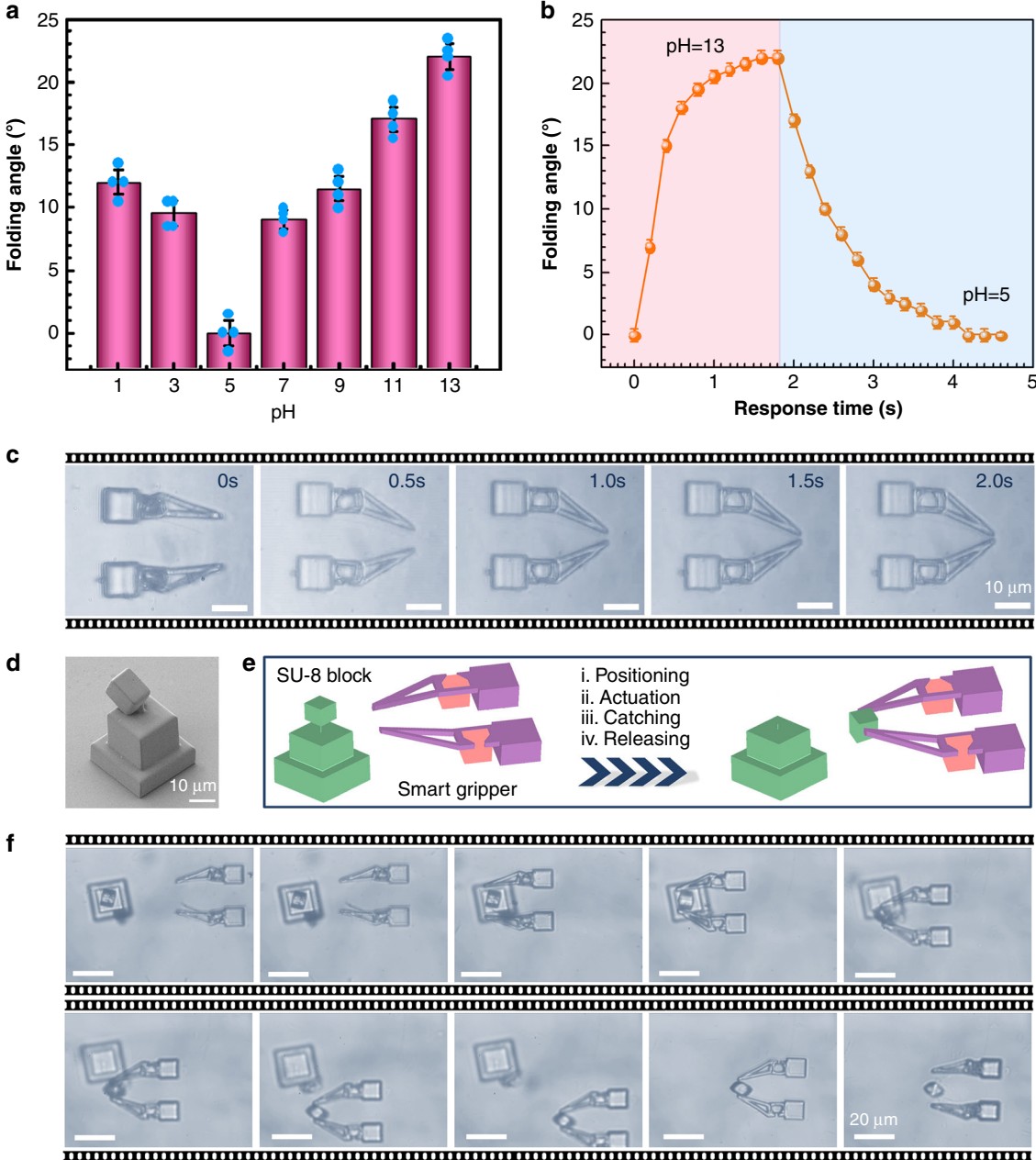

**Fig. 5 Flexible manipulation of the smart micro-gripper. a** The folding angle of the micro-grippers at different pH values. **b** The pH-responsive folding dynamics of the micro-grippers. **c** The actuation process of the micro-gripper in an aqueous solution with pH = 13. **d** SEM image of the target cargo, a SU-8 block attached to a base. **e** Schematic procedure of the catching and releasing of the SU-8 cargo using the pH-responsive micro-gripper. **f** The real process of the manipulation: (i) positioning, the micro-gripper that is integrated with a glass cantilever gradually approached the target and precisely aligned with it with the help of a 3D moving stage; (ii) actuation, the micro-gripper was stimulated by the solution with pH = 13, which triggered the folding of the micro-gripper. In this way, the target block was tightly gripped; (iii) transporting, the micro-gripper holding the object can be transferred to any desired location under the precise control of the motion system; (iv) releasing, the micro-gripper was stimulated by a solution of pH 5, which caused the unfolding of the arms, thus allowing the release of the object. Error bars denote the standard deviation of the measurements. Source data are provided as a Source data file.

the micro-gripper aligned with it, the pH of surrounding medium was switched to 13, triggering the actuation process. Due to the expansion of the BSA muscles, the target was tightly held by the micro-gripper. The object can be transported to any desired location under the precise control of the motion system. To accomplish the release, the solution pH value can be switched back to 5. The BSA muscles contract, and thus, the gripper opens, releasing the object.

Actually, in addition to our successive on-chip TPP fabrication, other micro/nanofabrication technologies also enable producing

stimuli-responsive micro-grippers. To provide the state-of-the-art of micro-grippers, methods for producing micro-grippers have been comprehensively summarized in Supplementary Table 1 (Supporting information). As compared to the existing studies, our micro-gripper possesses much smaller size (tens of micrometers), which gives rise to much more sensitive response due to the significantly increased surface-to-volume ratio. More importantly, unlike the bilayer grippers fabricated based on 2D patterns, our micro-gripper features 3D configurations by mimicking natural musculoskeletal systems, which makes it possible to

perform more complex deformations beyond gripping. The smart micro-gripper that enables controllable targeting, capturing, and releasing of desired micro-objects are promising for various cutting-edge applications. Especially, with the rapid progress of microfluidics[53–59], such robotic musculoskeletal systems may evolve to a robot-on-a-chip system and hold great promise for in vitro cell operation, cargo handling, precision assembly, device integration, and minimally invasive surgery.

## Discussion

In conclusion, we reported the femtosecond laser programmed artificial musculoskeletal systems for developing 3D microbots. A relatively stiff photopolymer, SU-8, was adopted as the building element of skeletons, and a pH-responsive soft material, BSA, was employed as muscles. To spatially program the distribution of SU-8 skeleton and BSA muscle at sub-micron scale during the TPP fabrication, we developed a successive on-chip TPP strategy that enables in situ TPP, development, photoresist exchange, and multiple TPP integration. By sequentially structuring of two photosensitive materials according to a preprogrammed model, smart microbots that permit stimuli-responsive actuation can be directly fabricated. Taking advantage of the direct writing feature of TPP, the internal network of both BSA and SU-8 can be programmed at nanoscale. In this way, the elasticity of the muscle and the stiffness of the skeleton can be tuned for flexible actuation. As a typical model, a pH-responsive spider microbot with BSA muscles integrated at the joints of SU-8 legs was successfully actuated. In addition, two typical musculoskeletal systems, the arm-muscle system and crab claw-muscle system, were fabricated for actuation and stability tests, which presented fast response, good durability and robustness after long-time storage (over 45 days). As a proof-of-concept application, a smart micro-gripper enabling pH-controlled capturing and releasing micro-targets was demonstrated. The successive on-chip TPP strategy that permits programmable integration of multiform materials into complex 3D microstructures is universal, not limited to the SU-8 and BSA system. More importantly, the strategy is compatible with other advanced micro/nanofabrication technologies that permit scalable and facile fabrication of any hierarchical structures, for instance, the soft replication process[60]. In combination with such cost-effective and scalable technologies, both the fabrication efficiency and uniformity of resultant devices would be promoted significantly. It is believable that, with the rapid development of smart materials and advanced processing technologies, successive TPP fabrication may find broad applications in prototyping 3D microbots that consist of multi-materials.

## Methods

**Preparation of photosensitive BSA hydrogel**. BSA (Sigma-Aldrich, A7638) and methylene blue (MB, Sigma-Aldrich, M9140) were dissolved in ultrapure water to form an aqueous gel with a concentration of 500 mg/mL for BSA and 0.6 mg/mL for MB. Then they were stored in a refrigerator at 4 °C for 24 h to allow complete dissolution of the reagents. The ultrapure water in this study (18.2 MΩ cm, 25 °C) was produced by using a water purification system (MILLIPORE).

**Fabrication of the SU-8 skeletons**. First, SU-8 2025 (NANO, MicroChem) was drop-casted onto the coverglass substrate equipped with PDMS parapets prepared by soft lithography. Then, SU-8 2025 photoresist was soft baked in an oven at 95 °C for 1 h to allow the sufficient evaporation of the solvents. The femtosecond laser beam (a repetition rate of 80 MHz, a pulse width of 120 fs, a central wavelength of 800 nm) was tightly focused into the SU-8 2025 photoresist by an oil-immersion objective lens (Olympus, ×60, numerical aperture = 1.4). The average laser power measured before the objective lens was ~10 mW. The focused laser spot scanned in the photoresist in three dimensions by the cooperative control of a two-galvano-mirror set for the horizontal scanning and a piezo stage (Physik Instrument) for the vertical scanning, respectively. The step length during laser scanning could be chosen as 100, 200, or 300 nm in three dimensions according to different requirements. After that, the samples were post baked for additional 30 min at 95 °C to solidify the exposed regions. Subsequently, in situ development was carried out by injecting acetone from the entrance of the PDMS parapet to dissolve the unexposed photoresist. The developer and unpolymerized photoresist were collected from the exit of the PDMS parapets.

**Integration of the BSA muscles**. After the TPP fabrication of SU-8 skeleton, the BSA aqueous gel was added to the substrate for the following TPP process. The step length of laser scanning could be chosen among 50, 100, 150, and 200 nm in three dimensions according to the desired elasticity of the BSA muscles. The average laser power measured before the objective lens was tuned to 20 mW. The in situ development and exchange of multi-materials allowed the precise integration of BSA muscles in the pre-fabricated SU-8 skeletons. (See Supplementary Experimental details for in situ alignment of BSA muscles with SU-8 skeleton.) After completing the integration of BSA muscles, ultrapure water was injected through the entrance of the PDMS parapet to dissolve the unpolymerized BSA gel. In this manner, a micro-robot composed of the SU-8 skeleton and BSA muscle was produced. For proper storage, the device can be kept in aqueous solutions at ~ 4 °C in a sterile environment, and the ultrapure water can be replaced every 3 days.

**Characterization**. SEM images were acquired by using a field emission SEM (JSM-7500F, JEOL), and the samples needed to be coated with a gold film (a thickness of ~3–5 nm) beforehand. The optical microscopy images were obtained using a Motic BA400 microscope equipped with a charge coupled device camera. The pH of the solutions was calibrated by using a precision pH meter PHS-25 (INESA) with a resolution of 0.01. The Young's modulus of the SU-8 microstructures was measured by using an Agilent Nano Indenter G200 equipped with an XP-style actuator, and the continuous stiffness measurement method was adopted. The measurement was carried out using a Berkovich diamond tip at 30 °C with a relative humidity of 20%.

## Data availability

The data that support the findings of this study are available from the corresponding author upon reasonable request. Source data are provided with this paper.

## Code availability

All the relevant code used to generate the results in this paper and Supplementary information is available upon request.

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

## Acknowledgements

This work was supported by the National Natural Science Foundation of China (NSFC) and the National Key Research and Development Program of China under Grant Nos. #61590930, #61935008, #61775078, #61905087 and #2017YFB1104300 and the China Postdoctoral Science Foundation under Grant 2020T130342.

## Author contributions

Z.-C.M., Y.-L.Z., and H.-B.S. conceived the idea and designed the research. Z.-C.M., B.H., X.-Y.H., and C.-H.L. undertook the experiments and the characterization. Q.-D.C. helps for setting up the laser fabrication system. Z.-C.M., Y.-L.Z., and H.-B.S. wrote the paper.

## Competing interests

The authors declare no competing interests.
