## [Peer Review File · Nature Communications]

Reviewers' comments:

Reviewer #1 (Remarks to the Author):

This is an attractive and impressive paper. Including reliable video demonstration, this work would attract a wide range of readers. I basically recommend publication of this nice work in Nature Communications. This well popular work would fit well with this journal. Technical contents of this research are mostly convincing. I have some requests for rather additional points.

1) Technical points such as robotic motions are well completed in this work. In contrast, very basic points were not well confirmed. At least, confirmation of used materials by IR spectroscopy or the other appropriate methods has to be done. Such data have to be added to Supporting Information.

2) Recent research activities on micro-motors for biomedical applications can be included to Introduction (for example, Bull. Chem. Soc. Jpn. 92, 1754-1758 (2019)).

3) The manuscript itself is well organized and well written. However, some paragraphs are uncomfortably long. Such long paragraphs would be harmful for non-specialist readers. Please divide too long paragraph into two or three paragraphs with comfortable lengths.

4) Video demonstrations are very good, but addition of text explanation more would be good for improve understandability.

Reviewer #2 (Remarks to the Author):

This study focuses on the employment of femtosecond laser programmed artificial musculoskeletal systems for prototyping 3D microbots. The general scope of the paper is well justified and of interest to a sufficiently broad audience. The key novelty of this work is to achieve programmable assembly of two or more materials of distinct properties into complex 3D micro/nanostructures at sub-micron scale. However, there are several issues that I believe need to be addressed before publishing this manuscript can be recommended.

(1) As stated by the authors, the interface of BSA muscle and SU-8 skeleton is precisely integrated. Indeed, the seamless integration plays a crucial role in the actuation performance of smart micro-gripper. I am curious about the bonding/connection mechanism between two materials. Is it noncovalent or covalent from the molecular-level perspective? It is important to provide sufficient evidence to substantiate it.

(2) Considering the relatively weak mechanical strength of BSA hydrogel, I am curious about the durability of the resulting musculoskeletal. Also, what's the maximum weight that BSA muscles can drive.

(3) In recent years, several techniques for engineering bioinspired micro-grippers have been demonstrated. The authors should clearly highlight the major difference from existing studies, especially from the aspects of fabrication, materials and functions (for example, feature size,

response time, gripping capability and so on)

(4) The authors did a good job in reviewing the state of the art in micro/nanofabrication technologies. It is important to mention a new soft replication process which allows for the scalable and facile fabrication of any hierarchical structures including the musculoskeletal system presented in this paper (Proceedings of National Academy of Sciences, 2019, 116, 23909). Actually, it is more convenient to replicate one single component using this cost-effective method.

Reviewer #3 (Remarks to the Author):

The article entitled “Femtosecond Laser Programmed Artificial Musculoskeletal Systems” by Ma et. al. describes the fabrication of a microfabricated actuating technique through the integration of a pH responsive BSA structure into epoxy crosslinked polymer microstructures. Multimaterial integration is important for fabrication of functional structures. This paper presents very relevant results in this direction.

The authors have carried out technically challenging work of integrating multiple materials into the same microstructure. The precise patterning of one material over the other with sub 10 μm resolution is highly challenging and the authors have addressed it by designing an on chip TPP process. The presented results regarding pH dependent actuation, its reproducibility validate the claims of the article. It is a well written and presented article that merits publication in Nat. Comm. Provided it addresses the following comments.

Major

1. on-chip TPP: Accurate two-photon patterning of multiple materials is challenging work; the authors claim to have accomplished this with the use of a PDMS parapet over the cover glass . To have an accurate understanding of the presented process a photo of the cover glass with PDMS parapet (for on-chip TPP) be included with the supporting information.
2. Multi-material pattern alignment: A description of how the alignment BSA crosslinked muscles were achieved accurately on the intended site on the SU8 spider should be included in the materials-methods section or in the supporting information .
3. Spider leg: Is the bottom of the spider legs attached or detached to the surface of the substrate? This aspect is not clear from the SEM images presented with the manuscript. If it is detached from the surface is it possible to include a close-up image of the tip of the legs in the supplementary information because their contact with the substrate is crucial in understanding the actuation seen in Figure 1(g)
4. Calculation of the tension exerted by the BSA (in article and supporting information): The mechanical properties of two-photon polymerized microstructures of any given material vary greatly

from the mechanical properties measured for a bulk film of the same material. So, there might be discrepancies in the elasticity of SU8 (4.4 GPa) used in calculating the tension exerted by BSA. Therefore, in the spirit of full disclosure it is important to provide a reference for the source of the value of elasticity of SU8 (4.4 GPa) in the supporting information. If you want to change the value to recalculate the tension, references should be provided for the modulus values adapted for the recalculation. Any changes in force calculation should be consistent with the simulations (recalculate if necessary).

Minor

5. Figure 1(b)-(c): The optical images in the insets give a clearer evidence of integration of BSA on SU8 due to the distinctive blue color of the dye included in BSA. Consider making the optical images bigger.

6. Figure 1(g): The superposition image is confusing because of the reddish appearance of the stack, consider reducing the opacity. Purple is a bad choice of color because it is too dark to see. Consider using complementary colors.

7. Figure 2(a): This figure presents a good demonstration of the chemical mechanism of the actuation. However, the different charges (+/-) appearing in the chemical structures is difficult to spot. Make better highlighting of the charges (consider making positive and negative symbols bigger)

8. The symbol °C is broken in many places in the document, please re-check every instance of °C in the document

9. Line 223: “approximately” should be changed to “approximate”; same change in the page 5 of supporting information.

Response to reviewers' comments

Reviewer #1:

This is an attractive and impressive paper. Including reliable video demonstration, this work would attract a wide range of readers. I basically recommend publication of this nice work in Nature Communications. This well popular work would fit well with this journal. Technical contents of this research are mostly convincing. I have some requests for rather additional points.

Response: Thanks a lot for the comments. We have carefully revised our manuscript, in which all of the comments have been suitably addressed. Changes have been highlighted in red, a point-by-point response has been listed as follow.

Comments 1: Technical points such as robotic motions are well completed in this work. In contrast, very basic points were not well confirmed. At least, confirmation of used materials by IR spectroscopy or the other appropriate methods has to be done. Such data have to be added to Supporting Information.

Response: Thanks a lot for the comments. This suggestion is quite helpful for improving our manuscript. Actually, our musculoskeletal systems mainly consist of SU-8 skeletons and BSA muscles. To confirm their chemical structures, Fourier Transform Infrared (FTIR) spectra of SU-8, BSA and their hybrid, have been added in the supporting information (Supplementary Fig. 4). Additionally, relative discussions have been included in the revised manuscript, which have been highlighted in red.

Page 6, Line 19: *“The used materials (SU-8, BSA and their hybrid) were characterized using Fourier Transform Infrared (FTIR) spectroscopy (Supplementary Fig. 4). Typically, the peaks at 3049 cm⁻¹, 1609 cm⁻¹, 758 cm⁻¹, and 659 cm⁻¹ in the spectrum of SU-8 (Supplementary Fig. 4a) correspond to the stretching vibration mode of aromatic =C-H, stretching vibration mode of aromatic C-C, out of plane bending mode of aromatic C-C, and out of plane bending mode of C-C. For BSA, the typical peaks at 3303 cm⁻¹, 1651 cm⁻¹, and 1536 cm⁻¹ correspond to the Amide A band (mainly N-H stretching vibration), Amide I band (stretching vibration of C=O), and*

Amide II band (stretching vibration of C-N and bending of N-H), respectively (Supplementary Fig. 4b).”

Supplementary Figure 4 | FTIR spectra of an SU-8 film (a), BSA film (b), and SU-8/BSA hybrid (c).

Comments 2: Recent research activities on micro-motors for biomedical applications can be included to Introduction (for example, *Bull. Chem. Soc. Jpn.* **92**, 1754-1758 (2019)).

Response: Thanks a lot for the comments and the recommended reference. Indeed, recent studies on micro-motors have revealed great potential for biomedical applications. To provide a comprehensive background, the micro-motors that enable self-propelled motion have been included to the introduction section, in which the above-mentioned reference has been suitably cited.

Page 3, Line 9: “Moreover, soft microbots *and micro-motors* can be actuated by a

wide variety of external stimuli (e.g., electricity, magnetic field, light, heat, pH value, humidity and chemical gradient), by which *self-propelled motion*, predictable deformation and controllable locomotion can be achieved without coupling additional energy-supply systems²²⁻³⁴. The aforementioned advantages make soft microbots promising for precise in-vivo/in-vitro operations, for instance, *human cell interaction*, targeted drug delivery and minimally invasive surgery^{2,4,5,22,35}.”

Ref 22 is the recommended reference (Bull. Chem. Soc. Jpn. 92, 1754-1758, 2019).

Comments 3: The manuscript itself is well organized and well written. However, some paragraphs are uncomfortably long. Such long paragraphs would be harmful for non-specialist readers. Please divide too long paragraph into two or three paragraphs with comfortable lengths.

Response: Thanks a lot for the valuable suggestion. We have revised our manuscript accordingly. Some long paragraphs have been divided into short paragraphs with suitable lengths.

Page 7, Line 17: The long paragraph is divided into two paragraphs.

Page 9, Line 10: The long paragraph is divided into two paragraphs

Page 10, Line 21: The long paragraph is divided into two paragraphs.

Comments 4: Video demonstrations are very good, but addition of text explanation more would be good for improve understandability.

Response: Thanks a lot for the valuable comments. The suggestion is quite useful for our manuscript. We have added subtitles explanation in every video. Besides, more explanations have been provided in the video captions (supporting information).

Supporting information, Page 18, Line 1:

“Movie captions:

Supplementary Movie 1. Actuation of the spider microbot (the pH value was switched from 13 to 5). Initially, the eight legs were bended since the BSA muscles were expanded at pH 13. When the pH value was switched to 5, the BSA muscles at the joints contracted, causing the straightening of the legs.

Supplementary Movie 2. *Dynamic actuation performance of the arm-muscle system (the surrounding pH value was switched from 5 to 13). At the beginning, the arm was straight since the BSA muscle was contracted (pH=5). The BSA muscle expanded once the pH value was switched to 13, which caused the folding of the arm.*

Supplementary Movie 3. *Dynamic actuation of the crab claw-muscle system. Initially, the crab claw was open since the BSA muscle was contracted when the pH value was 5. After the pH value was switched to 13, the BSA muscle expanded, leading to the closing of the claw.*

Supplementary Movie 4. *A 3D smart micro-gripper that consists of the SU-8 skeleton and BSA muscle. Initially, the gripper was open since the BSA muscle was contracted (pH=5). Then the BSA muscle expanded once the pH value was switched to 13, which caused the gripping motion of the gripper.*

Supplementary Movie 5. *Targeted capturing, transport, and releasing of a cargo using the smart micro-gripper. First, the micro-gripper moved to the location of the target (a micro-cube with a side length of 10 μm). Then the pH value of the surrounding medium was switched to 13, which leads to the expansion of BSA muscles and the gripping performance. Subsequently, the cube was held and transported to the desired location. Finally, the pH value was switched to 5, and the gripper released the cube due to the contraction of the BSA muscles.”*

Reviewer #2:

This study focuses on the employment of femtosecond laser programmed artificial musculoskeletal systems for prototyping 3D microbots. The general scope of the paper is well justified and of interest to a sufficiently broad audience. The key novelty of this work is to achieve programmable assembly of two or more materials of distinct properties into complex 3D micro/nanostructures at sub-micron scale. However, there are several issues that I believe need to be addressed before publishing this manuscript can be recommended.

Response: Thanks a lot for the comments. We have carefully revised our manuscript, in which all of the comments have been suitably addressed. Changes have been highlighted in red, a point-by-point response has been listed as follow.

Comments 1: As stated by the authors, the interface of BSA muscle and SU-8 skeleton is precisely integrated. Indeed, the seamless integration plays a crucial role in the actuation performance of smart micro-gripper. I am curious about the bonding/connection mechanism between two materials. Is it noncovalent or covalent from the molecular-level perspective? It is important to provide sufficient evidence to substantiate it.

Response: Thanks a lot for the comments. Indeed, the seamless adhesion of two materials is essential for actuation. In our work, the SU-8 structure was fabricated in the first run of TPP process, including soft baking, photopolymerization, post baking and development. In principle, the SU-8 structure has been already solidified. Therefore, the polymerization of BSA in the second run of TPP process would not trigger covalent bonding with the inert SU-8 surface, since the two materials polymerized through different ways. To confirm this issue, the FTIR spectrum of the SU-8/BSA hybrid was carefully compared with pure SU-8 and BSA. As shown in Supplementary Fig. 4, there are no new vibration peaks corresponding to the newly formed covalent bonds. Similar with the interaction between SU-8 and the glass substrate, the BSA muscle interacts with SU-8 through a noncovalent manner, mainly due to the Van der Waals force. Nevertheless, seamless integration of the two

materials can be guaranteed, since the BSA and SU-8 voxels overlap each other in our laser scanning program. To clarify this interfacial interaction, the FTIR spectra and relative discussions have been added in the revised manuscript.

Page 8, Line 14: *“The seamless integration of BSA and SU-8 is essential for the actuation performance. In principle, the two materials polymerize through different mechanisms. Thus, the BSA muscle may interact with SU-8 through a noncovalent manner. The FTIR spectrum of the SU-8/BSA hybrid (Supplementary Fig. 4) confirmed this issue, in which vibration peaks corresponding to newly formed covalent bonds cannot be detected.”*

Supplementary Figure 4 | FTIR spectra of an SU-8 film (a), BSA film (b), and SU-8/BSA hybrid (c).

Comments 2: Considering the relatively weak mechanical strength of BSA hydrogel, I am curious about the durability of the resulting musculoskeletal. Also, what's the maximum weight that BSA muscles can drive.

Response: Thanks a lot for the comments. Actually, our musculoskeletal systems possess good durability after long time storage under a suitable condition. The SEM images of the arm-muscle structure after storage at 25 °C for 45 days may confirm this argument (Fig. 3a, v-vi). Notably, after 45 days, the arm-muscle device almost keeps the same morphology and comparable performance as the pristine one, indicating the good stability. Generally, the good durability can be attributed to the following two points. First, the stiff SU-8 structure can act as the skeleton to support the whole musculoskeletal systems, so the device is robust enough for repeated actuation. Second, both SU-8 and the BSA hydrogel are chemically stable under proper storage conditions. The skeleton material SU-8 is a chemically inert resin that is very stable at room temperature, whereas the BSA muscle should be preserved in ultrapure water at low temperature to avoid biodegradation. To address the concerns with respect to the durability, more information about the proper storage method has been added in the revised manuscript. Briefly, the device can be kept in aqueous solution at ~ 4 °C in a sterile environment. Besides, the ultrapure water can be replaced every three or four days.

The maximum weight that BSA muscles can drive depends on the size of BSA, the pH value for actuation and the laser processing parameters (*e.g.*, the laser scanning step length, the laser power). Therefore, to analyze the maximum weight driven by the BSA muscle, we fixed the physical parameters of the BSA muscle model (25 μm in length and 2 μm in width, laser scanning step length of 100 nm, laser power of 20 mW) and varied the widths of the SU-8 cantilevers within a certain range to test the driving force (4 μm, 6 μm, and 8 μm, Supplementary Fig. 9). According to the experimental results, the tip displacement of the SU-8 cantilevers with a width of 4 μm and 6 μm was measured to be 1.5 μm and 0.6 μm, respectively (Supplementary Fig. 9a-d). Combining with the COMSOL Multiphysics simulation, the exerted force at the tip of the cantilever with a width of 4 μm was calculated to be ~17 μN

(Supplementary Fig. 9g). In the case of SU-8 cantilever with a width of 6 μm , the exerted force at the tip was calculated to be $\sim 34 \mu\text{N}$ (Supplementary Fig. 9h). This value can be considered as the maximum weight that the BSA muscle can drive, because it cannot drive the SU-8 cantilever with a larger size (e.g., 8 μm in width, Supplementary Fig. 9e and 9f). The experimental results and relative discussions have been added in the revised manuscript, which have been highlighted in red.

Page 9, Line 3: “*Generally, the good durability can be attributed to the following two points. First, the stiff SU-8 structure can serve as the skeleton to support the whole musculoskeletal systems, so the device is robust enough for repeated actuation. Second, both SU-8 and the BSA hydrogel are chemically stable under proper storage conditions. The skeleton material SU-8 is a chemically inert resin that is very stable at room temperature, whereas the active muscle material BSA should be preserved in ultrapure water at low temperature to avoid biodegradation.*”

Page 17, Line 1, Methods section: “*For proper storage, the device can be kept in aqueous solutions at $\sim 4 \text{ }^\circ\text{C}$ in a sterile environment, and the ultrapure water can be replaced every three days.*”

Page 12, Line 4: “*The maximum weight that BSA muscles can drive depends on the size of BSA, the pH value for actuation and the laser processing parameters. To analyze the maximum weight driven by the BSA muscle, we fixed the physical parameters of the BSA muscle model (25 μm in length and 2 μm in width, laser scanning step length of 100 nm, laser power of 20 mW) and varied the widths of the SU-8 cantilevers within a certain range (4 μm , 6 μm , and 8 μm , Supplementary Fig. 9). According to the experimental results, the BSA muscle can drive the SU-8 cantilevers with a width of 6 μm , corresponding to the maximum force of $\sim 34 \mu\text{N}$. The SU-8 cantilever of 8 μm in width cannot be driven.*”

Supplementary Figure 9 | Experimental analysis and COMSOL simulation of the displacement of SU-8 cantilevers with different widths (4 μm, 6 μm, and 8 μm). a, b Optical microscopy images of the SU-8 cantilever with a width of 4 μm before and after integration of a BSA muscle at the free end. **c, d** Optical microscopy images of the SU-8 cantilever with a width of 6 μm before and after integration of a BSA muscle at the free end. **e, f** Optical microscopy images of the SU-8 cantilever with a width of 8 μm before and after integration of a BSA muscle at the free end. **g**, Simulation of the tip displacement of the SU-8 cantilever with a width of 4 μm when the exerted force on the tip is 17 μN. **h**, Simulation of the tip displacement of the SU-8 cantilever with a width of 6 μm when the exerted force on the tip is 34 μN.

Comments 3: In recent years, several techniques for engineering bioinspired micro-grippers have been demonstrated. The authors should clearly highlight the major difference from existing studies, especially from the aspects of fabrication, materials and functions (for example, feature size, response time, gripping capability and so on).

Response: Thanks a lot for the comments. To provide the state-of-the-art of micro-grippers, the fabrication methods, used materials, size of devices, response time, actuation mechanisms, and the gripping capabilities of previously reported micro-grippers have been comprehensively summarized in a table (Supplementary Table 1). As compared to the existing studies, our micro-gripper features much smaller sizes (tens of micrometers), faster response (response time as short as hundreds of milliseconds) and essentially 3D configurations. More importantly, unlike the bilayer grippers fabricated based on 2D patterns, our micro-gripper is the first one that mimics natural musculoskeletal systems, which makes it possible to perform more complex deformations beyond gripping, holding great promise for developing 3D microbots. To make our micro-gripper distinguished with others, the table has been added to supporting information (Supplementary Table 1), and relative discussions have been added to the revised manuscript.

Page 13, Line 14:

“Actually, in addition to our successive on-chip TPP fabrication, other micro/nanofabrication technologies also enable producing stimuli responsive micro-grippers. To provide the state-of-the-art of micro-grippers, methods for producing micro-grippers have been comprehensively summarized in Supplementary Table 1 (Supporting information). As compared to the existing studies, our micro-gripper possesses much smaller size (tens of micrometers), which gives rise to much more sensitive response due to the significantly increased surface-to-volume ratio. More importantly, unlike the bilayer grippers fabricated based on 2D patterns, our micro-gripper features 3D configurations by mimicking natural musculoskeletal systems, which makes it possible to perform more complex deformations beyond gripping.”

Supplementary Table 1 | The state-of-the-art of the reported micro-grippers.

Fabrication techniques	Materials	Size of devices	Actuation mechanism	Response time	Gripping capability	Ref.
successful on-chip TPP strategy	SU-8/BSA	40 μm \times 35 μm	pH	~1 s	Pick and place micro-object (0–20 μm)	This work
Dual-3D femtosecond laser nanofabrication	PBMA	D=40 μm	Acetone/hexane	~1.5 s	Pick and place microsphere of ~ 20 μm	1
photolithography	PCL/ PNIPAM bilayers ^a	D=450 μm	Temperature	10 s	encapsulate and release yeast cells	2
photolithography, electrodeposition, and etching	Cr/Au bilayer, SC1805 and SC1813	D= ~ 980 μm	Temperature	10 min	tissue excision and biopsy of the bile duct	3
photolithography	PPF/ pNIPAM-AAc ^b	D= ~1 mm	Temperature	11 min	grip onto tissue and elute a drug	4
photolithography	POEGMA ^c	D= ~6 mm in	temperature	10 min	four-state shape changes	5
EBL ^d , electron-beam evaporation and lift-off	SiO ₂ and Fe	250 μm in length and 170 μm in width	Magnetic field	-	grip a single cell	6
photolithography	Au/Ni, Cr/ Cu/ novolac resin)	D= 700 μm	acetic acid and hydrogen peroxide	30 s	Pick and place beads (D=200 μm)	7
Photolithography, electroplating, and etching	Cr/Cu bilayer	D= ~ 1.2 mm	O ₂ and H ₂	25 s	bidirectional gripping	8
photolithography	Gelatin and CMC ^e	D= ~1.1 mm	enzyme	-	grip and release beads (D=700 μm)	9
Two-photon polymerization	Liquid crystal	D= ~200 μm	Light	0.1 s	Catch microcubes (40 \times 40 \times 20 μm^3)	10
photolithography	cresol novolac resin and a Cr/Cu thin film	D= ~700 μm	Temperature, glucose, and trypsin	40 s	removal of cells from tissue	11
photolithography	PEGDA ^f /NIPAAm bilayer with magnetic alginate microbeads	D= ~400 μm	Temperature, light and magnetic field	-	release of microparticles	12

photolithography	PHEMA ^g and PEGDA with Fe ₃ O ₄ NPs	D= ~2 mm	Magnetic field and pH	1.5 min	Trap and release beads (D=300 μm)	13
Photolithography and thermal evaporation	SU-8/gold layer	2~3 mm long and 300~400 μm wide	Electrothermal	-	Cell manipulation	14
SOI process ^h	silicon	2900× 2700 μm	Electrostatic	-	Micro-object manipulation	15
μEDM ⁱ , PECVD ^j , and etching	Shape memory alloy (Ti/Ni) and SiO ₂	33 mm × 9 mm × 3 mm	Electrothermal	7.5~9 s	Micro-object manipulation	16
MEMS processes (LPCVD, sputtering, and ICP-RIE)	sol-gel multi-coated PZT (Pb(Zr _{0.52} Ti _{0.48})O ₃) films	2 mm-long and 100 μm wide	Piezoelectric	-	grip metallic balls (D= 100 μm)	17
electro-discharge machining	superelastic alloy (NiTi)	length 15.5 mm, width 5.22 mm, thickness 0.5 mm	Electromagnetic	-	tissues manipulation (~300 μm)	18
Photolithography and replica molding	polyurethane, magnetic microparticles, and hollow glass microbeads	700 μm in length and 70 μm in width	Magnetic field	-	Pick and place microgels	19
Two-photon polymerization	IP-Dip photoresist	~100 μm in length and breadth	External force applied to the center of the gripper	-	grasp, move and release objects (50× 50 × 70 μm ³)	20
Photolithography and e-beam evaporation	SiO/SiO ₂ bilayer	D= ~ 70 μm	residual stress within the prestressed bilayer	-	capture of live mouse cells	21

D: diameter

NPs: nanoparticles

^a PCL: polycaprolactone

PNIPAM: poly-(N-isopropylacrylamide)

^b PPF: poly(propylene fumarate)

pNIPAM-AAc: poly(N-isopropylacrylamide-co-acrylic acid)

^c POEGMA: poly[oligo (ethylene glycol) methyl ether methacrylate]

^d EBL: electron-beam lithography

^e CMC: carboxymethylcellulose

^f PEGDA: poly (ethylene glycol) acrylate

^g PHEMA: 2-hydroxyethyl methacrylate

^h SOI: silicon on insulator

ⁱ μ EDM: micro-electrical-discharge-machining

^j PECVD: plasma-enhanced chemical vapor deposition

The detailed reference list can be seen in the revised supporting information.

Comments 4: The authors did a good job in reviewing the state of the art in micro/nanofabrication technologies. It is important to mention a new soft replication process which allows for the scalable and facile fabrication of any hierarchical structures including the musculoskeletal system presented in this paper (Proceedings of National Academy of Sciences, 2019, 116, 23909). Actually, it is more convenient to replicate one single component using this cost-effective method.

Response: Thanks a lot for the comments and the recommended reference. Indeed, the soft replication technique that enables scalable and facile fabrication of hierarchical micro/nanostructures is of great significance to the development of microbots. Especially, it is compatible with our successive TPP fabrication. Providing one of the components could be fabricated *via* such a cost-effective and scalable process, both the fabrication efficiency and the uniformity of resultant devices would be promoted significantly. To provide a prospect of the future, we introduced the unique merits of soft replication process and its potential contribution to our strategy in the conclusion section, in which the recommended reference has been suitably cited.

Page 14, Line 22: “More importantly, the strategy is compatible with other advanced micro/nanofabrication technologies that permit scalable and facile fabrication of any hierarchical structures, for instance, the soft replication process⁶⁰. In combination with such cost-effective and scalable technologies, both the fabrication efficiency and the uniformity of resultant devices would be promoted significantly.”

60. Li W, et al. Crack engineering for the construction of arbitrary hierarchical architectures. *Proc Natl Acad Sci U S A* 116, 23909-23914 (2019).

Reviewer #3:

The article entitled “Femtosecond Laser Programmed Artificial Musculoskeletal Systems” by Ma et. al. describes the fabrication of a microfabricated actuating technique through the integration of a pH responsive BSA structure into epoxy crosslinked polymer microstructures. Multimaterial integration is important for fabrication of functional structures. This paper presents very relevant results in this direction.

The authors have carried out technically challenging work of integrating multiple materials into the same microstructure. The precise patterning of one material over the other with sub 10 μm resolution is highly challenging and the authors have addressed it by designing an on chip TPP process. The presented results regarding pH dependent actuation, its reproducibility validate the claims of the article. It is a well written and presented article that merits publication in Nat. Comm. Provided it addresses the following comments.

Response: Thanks a lot for the comments. We have carefully revised our manuscript, in which all of the comments have been suitably addressed. Changes have been highlighted in red, a point-by-point response has been listed as follow.

Comments 1: on-chip TPP: Accurate two-photon patterning of multiple materials is challenging work; the authors claim to have accomplished this with the use of a PDMS parapet over the cover glass. To have an accurate understanding of the presented process a photo of the cover glass with PDMS parapet (for on-chip TPP) be included with the supporting information.

Response: Thanks a lot for the comments. The photograph of the PDMS parapet for the on-chip TPP has been added in the supporting information (Supplementary Fig. 1). The parapet consists of a two inlet channels, for the injection of photoresist and developer, a PDMS chamber for TPP fabrication and developing, as well as one outlet channel for waste discharge.

Supplementary Figure 1 | The photograph of the cover glass with a PDMS parapat for on-chip TPP. The parapat consists of a two inlet channels, for the injection of photoresist and developer, a PDMS chamber for TPP fabrication and developing, as well as one outlet channel for waste discharge.

Comments 2: Multi-material pattern alignment: A description of how the alignment BSA crosslinked muscles were achieved accurately on the intended site on the SU8 spider should be included in the materials-methods section or in the supporting information.

Response: Thanks a lot for the comments. A detailed description about the precise alignment and integration of BSA muscles on the intended site of the SU-8 spider has been provided. Briefly, the precise alignment of BSA with the SU-8 skeleton relies on the laser scanning program that has been divided into two complementary parts. Since the program of the BSA muscles was well complementary to that of the SU-8 skeleton, re-positioning process is not necessary. The as-formed BSA voxels overlap well with the SU-8 skeleton at the vacancies. In this manner the muscles can be aligned accurately at the desired sites on the SU-8 spider. To make the method clear, we added additional description of the experimental details in the supporting information, which has been marked in red.

Supporting information, Page 4, Line 1:

“Experimental details for in-situ alignment of BSA muscles with SU-8 skeleton.

To realize in-situ integration of BSA muscles with SU-8 skeleton, the whole program of the musculoskeletal spider was divided into two complementary parts: skeleton and muscle. In the first run, a standard TPP process was carried out for the polymerization of the SU-8 spider skeleton with eight muscle vacancies. After that, the laser scanning program was suspended. Then in-situ development process was performed. As a developer, acetone was injected into the PDMS chamber to remove the unpolymerized photoresist. Subsequently, BSA gel was injected into the chamber for the second run of TPP fabrication. After that, the laser scanning program was restarted to continue with the laser scanning process of the BSA muscles. Since the program of the BSA muscles was well complementary to that of the SU-8 skeleton, re-positioning process is not necessary. The as-formed BSA voxels overlap well with the SU-8 skeleton at the vacancies. In this manner the muscles can be aligned accurately at the desired sites on the SU-8 spider.”

Comments 3: Spider leg: Is the bottom of the spider legs attached or detached to the surface of the substrate? This aspect is not clear from the SEM images presented with the manuscript. If it is detached from the surface is it possible to include a close-up image of the tip of the legs in the supplementary information because their contact with the substrate is crucial in understanding the actuation seen in Figure 1(g)

Response: Thanks a lot for the comments. Actually, the bottom of the spider legs is detached to the substrate, as shown in the 3D model (Supplementary Fig. 3). Only in this way can the legs be reversibly actuated between bending and straightening by varying the pH values, this is also confirmed by Supplementary Movie 1. Nevertheless, for SEM characterization, the sample is treated in high vacuum environment, the suspended thin legs are prone to shrink and get close to the substrate. To make it clear, the 3D model and close-up SEM images of the tip of the legs have been added in the supporting information (Supplementary Fig. 3), and relative discussions have also been added in the revised manuscript.

Page 6, Line 9: *“Actually, the bottom of the spider legs is detached to the surface of*

the substrate, as shown in the 3D model and the close-up SEM images (Supplementary Fig. 3). In aqueous solutions, these legs can be driven by varying the pH values of the surrounding medium.”

Supplementary Figure 3 | A 3D model of the spider microbot and the SEM images of the tip of the legs. a, Side view of the 3D model. b-e, Close-up SEM images of the tips of the legs.

Comments 4: Calculation of the tension exerted by the BSA (in article and supporting information): The mechanical properties of two-photon polymerized microstructures of any given material vary greatly from the mechanical properties measured for a bulk film of the same material. So, there might be discrepancies in the elasticity of SU8 (4.4 GPa) used in calculating the tension exerted by BSA. Therefore, in the spirit of full disclosure it is important to provide a reference for the source of the value of elasticity of SU8 (4.4 GPa) in the supporting information. If you want to change the value to recalculate the tension, references should be provided for the modulus values adapted for the recalculation. Any changes in force calculation should be consistent with the simulations (recalculate if necessary).

Response: Thanks a lot for the valuable comments. In the previous version of our manuscript, the elasticity of SU-8 (4.4 GPa) came from the reference (J. Micromech. Microeng. 14 (2004) 1576–1584). Indeed, we agree that the mechanical properties of two-photon polymerized microstructures may vary greatly from that of a bulk film of the same material. Consequently, we measured the Young’s modulus of two-photon

polymerized SU-8 microstructures (cuboid: $200\ \mu\text{m} \times 200\ \mu\text{m} \times 6\ \mu\text{m}$, fabricated under the same condition) for recalculation. According to the experimental results from nanoindentation, the Young's modulus (E) is $\sim 4.8\ \text{GPa}$, only slightly higher than $4.4\ \text{GPa}$, as shown in Supplementary Fig. 6. Then, we recalculate the tension and also renew the simulation using this experimental value. The recalculated force values are also consistent with the simulations. For instance, the previously calculated tension was $1.3\sim 1.4\ \mu\text{N}$, and the recalculated tension is slightly higher, $1.4\sim 1.5\ \mu\text{N}$. The measurement of Young's modulus, recalculation procedure, and the corresponding results have been added in the revised manuscript and supporting information, which have been marked in red.

Supplementary Figure 6 | Young's modulus of SU-8 microstructures. The measurement was carried out at $30\ ^\circ\text{C}$ with a relative humidity of 20%.

Article, Page 17, Line 7: “*The Young's modulus of the SU-8 microstructures was measured by using an Agilent Nano Indenter G200 equipped with an XP-style actuator, and the continuous stiffness measurement method was adopted. The measurement was carried out using a Berkovich diamond tip at $30\ ^\circ\text{C}$ with a relative humidity of 20%.*”

Article, Page 11, Line 11: “*The Young's modulus of the SU-8 microstructures fabricated under this condition was measured to be $4.8\ \text{GPa}$ (Supplementary Fig. 6).*”

Supporting information, Page 12, Line 10: “ E is the modulus of elasticity of the cantilever material SU-8 (4.8 GPa), $I=(2\ \mu\text{m})^4/12$. If the tension $F= 1.4\ \mu\text{N}$, then $w= -1.96\ \mu\text{m}$; if $F= 1.5\ \mu\text{N}$, $w= -2.10\ \mu\text{m}$the tension of the BSA muscle (with a width of $2\ \mu\text{m}$ and a length of $25\ \mu\text{m}$) on the SU-8 cantilever is about $1.4\sim 1.5\ \mu\text{N}$.”

Supplementary Figure 8 | Simulation of the displacement of an SU-8 cantilever using COMSOL Multiphysics.

Supporting information, Page 14, Line 3: “The cantilever is linear elastic with Young's modulus $E= 4.8\ \text{GPa}$, ... When the force F is set to be $1.4\sim 1.5\ \mu\text{N}$, the tip displacement of the SU-8 cantilever is $\sim 1.98\sim 2.12\ \mu\text{m}$.”

Article, Page 11, Line 19: “The tension was calculated to be $1.4\sim 1.5\ \mu\text{N}$... When the applied force F is $1.4\sim 1.5\ \mu\text{N}$, the deflection of the end is $1.98\sim 2.12\ \mu\text{m}$, ...produce a tensile force of $1.4\sim 1.5\ \mu\text{N}$ on the SU-8 cantilevers upon pH change.”

Comments 5: Figure 1(b)-(c): The optical images in the insets give a clearer evidence of integration of BSA on SU8 due to the distinctive blue color of the dye included in BSA. Consider making the optical images bigger.

Response: Thanks a lot for the helpful suggestion. We have re-edited Figure 1, in which the optical images have been enlarged, as shown below.

Comments 6: Figure 1(g): The superposition image is confusing because of the reddish appearance of the stack, consider reducing the opacity. Purple is a bad choice of color because it is too dark to see. Consider using complementary colors.

Response: Thanks a lot for the valuable suggestion. To make the superposition image clearer, we have adjusted the opacity and the purple color has also been changed. In the new Figure 1, red, yellow and green are used to increase the contrast. Hence Figure 1 has been updated in the revised manuscript, which can be seen as follow.

Comments 7: Figure 2(a): This figure presents a good demonstration of the chemical mechanism of the actuation. However, the different charges (+/-) appearing in the chemical structures is difficult to spot. Make better highlighting of the charges (consider making positive and negative symbols bigger)

Response: Thanks a lot for the helpful suggestion. To make better highlighting of the charges, all the positive and negative symbols in Figure 2a have been enlarged, as shown below.

Comments 8: The symbol °C is broken in many places in the document, please re-check every instance of °C in the document

Response: Thanks a lot for the comments. We have double-checked every instance of °C throughout the manuscript and corrected all the broken symbols.

Page 8, Line 21: "... after storage at 25 °C for 45 days..."

Page 16, Line 5: "... 4 °C for 24 hours..."

Page 16, Line 6: "... (18.2 MΩ cm, 25 °C)..."

Page 16, Line 9: "... was soft baked in an oven at 95 °C for 1 h..."

Page 16, Line 17: "... the samples were post baked for additional 30 min at 95 °C..."

Comments 9: Line 223: "approximately" should be changed to "approximate"; same change in the page 5 of supporting information.

Response: Thanks a lot for the comments. We have changed all these typos accordingly.

Article, Page 11, Line 17: "Accordingly, a mathematical model based on *approximate* differential equation..."

Supporting information, Page 10, Line 3: "The tension of the BSA muscle can be calculated according to the classical *approximate* differential equation..."

REVIEWERS' COMMENTS:

Reviewer #1 (Remarks to the Author):

Replies and revisions are fine. The revised version becomes acceptable.

Reviewer #2 (Remarks to the Author):

I appreciated the authors' efforts in addressing all the comments raised by us. The revised paper is now technically solid and intellectually novel to warrant its publication in Nature Communications.

Reviewer #3 (Remarks to the Author):

Authors have satisfactorily addressed my concerns with the manuscript with this revision. I recommend acceptance of the manuscript.

Response to reviewers' comments

REVIEWERS' COMMENTS:

Reviewer #1 (Remarks to the Author):

Replies and revisions are fine. The revised version becomes acceptable.

Response: Thank you very much for your time and positive comments.

Reviewer #2 (Remarks to the Author):

I appreciated the authors' efforts in addressing all the comments raised by us. The revised paper is now technically solid and intellectually novel to warrant its publication in Nature Communications.

Response: Thank you very much for your time and positive comments.

Reviewer #3 (Remarks to the Author):

Authors have satisfactorily addressed my concerns with the manuscript with this revision. I recommend acceptance of the manuscript.

Response: Thank you very much for your time and positive comments.